# Proposing Clinicopathological Staging and Mitotic Count as Prognostic Factors for Canine Soft Tissue Sarcomas

**DOI:** 10.3390/vetsci10050327

**Published:** 2023-05-02

**Authors:** Andrea Regina Cardoso de Almeida Moreira, Mayara Simão Franzoni, Denner Santos dos Anjos, Paulo César-Jark, Juliano Nóbrega, Renée Laufer-Amorim, Marina Valsecchi Henriques, Osmar Pinto Neto, Carlos Eduardo Fonseca-Alves

**Affiliations:** 1Department of Veterinary Clinic, School of Veterinary Medicine and Science, São Paulo State University (UNESP), Botucatu 18618-681, Brazil; 2Eletro-Onkovet Service, Franca 14406-005, Brazil; 3Onccarevet Clinic, Ribeirão Preto 14026-587, Brazil; 4Biomedical Engineering Department, Anhembi University, São Paulo 04546-001, Brazil; 5Arena235 Research Lab, São José dos Campos 12246-876, Brazil; 6Institute of Health Sciences, Paulista University (UNIP), Bauru 17048-290, Brazil; 7Veterinary Oncology Service (SEOVET), São Paulo 05016-000, Brazil

**Keywords:** canine, mesenchymal tissues, sarcomas, overall survival

## Abstract

**Simple Summary:**

Soft tissue sarcomas (STSs) comprise different tumor subtypes with similar clinical and pathological characteristics that arise from the embryonic mesoderm. They affect both humans and dogs and are located in the cutaneous or subcutaneous tissue of different body structures. In humans, this tumor group has been well-studied and described, and several clinical and pathological factors are used to evaluate patient prognosis, including a clinicopathological unified staging system. However, in dogs, no staging system has been proposed for these tumors. Therefore, we adapted human clinicopathological staging for STS and applied it to a large group of dogs affected by STS. The staging system proposed ranged from stage I to IV and was able to separate patient prognosis, with stage I patients having a higher survival time and stage IV patients having the lowest survival time. Moreover, we investigated different clinical and pathological factors associated with prognosis in this set of patients. We then evaluated the median value, which was five for the mitotic count. Therefore, we evaluated whether this value could separate patient survival. It was possible to identify a higher survival time in patients with a mitotic count ≤5 than in those with a count >5.

**Abstract:**

Soft tissue sarcomas (STSs) are a heterogeneous group of malignant mesenchymal tumors with similar histological features and biological behaviors. They are characterized by a low to moderate local recurrence rate and low metastasis, affecting approximately 20% of patients. Although this tumor set is vital in veterinary medicine, no previous unified staging system or mitotic count has been associated with patient prognosis. Therefore, this study proposed a new clinicopathological staging method and evaluated a cut-off value for mitosis related to the survival of dogs affected by STS. This study included 105 dogs affected by STS, treated only with surgery, and a complete follow-up evaluation. The new clinicopathological staging system evaluated tumor size (T), nodal involvement (N), distant metastasis (M), and histological grading criteria (G) to categorize the tumor stage into four groups (stages I, II, III, and IV). The proposed tumor staging system was able to differentiate patients’ prognoses, with dogs with stage IV disease experiencing the lowest survival time and dogs with stage I disease having the highest survival time (*p* < 0.001). Moreover, we assessed the median mitosis (based on mitotic count) and its association with overall survival. Our study’s median mitosis was 5, and patients with ≤5 mitoses had a higher survival time (*p* = 0.006). Overall, the proposed staging system and mitotic count seemed promising in the prediction of patient prognosis.

## 1. Introduction

Soft tissue sarcomas (STSs) are a heterogeneous group of malignant tumors originating from mesenchymal cells of connective tissues and comprise a group with similar microscopic and clinical features [1,2]. STS occurs at any anatomical site; however, it commonly involves the cutaneous and subcutaneous tissues, accounting for 8–20% of all tumors at this site [1,3,4]. STS may originate from different cells, including adipocytes, neuronal, muscle, fascial, blood vessel walls, and fibroblasts [2]. Middle-to-older-aged dogs are more commonly affected, without sexual and breed predilection. Metastasis is hematogenous, affecting approximately 20% of patients. Lymph nodes are generally unaffected, and there is usually significant local infiltration [2,5].

Fine needle aspiration cytology is a screening method in patients with cutaneous nodules, but it can be inconclusive in cases of STS. The final diagnosis is usually made by histopathological analysis, preferably using tissue samples obtained by incisional biopsy, which allows the evaluation of the type of tumor and its grading [1,2,3,4,5]. The histopathological analysis of these tumors could represent a diagnostic challenge, due to the presence of similar cellular patterns among them, and a variety of other neoplasms with different histogenesis. The development of immunohistochemical techniques and the availability of monoclonal and polyclonal antibodies representing various tissue markers and tissue microenvironments have been of paramount importance in improving the diagnosis of STS [1,2,3].

The veterinary literature lacks information related to the standardization of this tumor group and a significant amount of the previously published articles are isolated case reports [6,7] or articles investigating different aspects of one specific histological subtype and not using these tumor groups as a unit [8,9]. As opposed to dogs, in humans with STS, a more standardized approach is reported for patient staging [10] and treatment [11]. For a better comprehension of this tumor group, different initiatives were created for a better patient approach by the American Joint Committee on Cancer (AJCC) and European Society for Medical Oncology [10,11]. For this reason, new strategies should be made in veterinary oncology for a better approach to managing canine STS.

Usually, canine STS presents as solitary and pseudo-encapsulated masses with poorly defined margins [5,12,13,14]. The histologic evaluation of these tumors is challenging because of similar cell patterns to various other neoplasms with different histogenesis [15,16]. In human medicine, STSs from the extremities and trunk are considered a group of tumors with similar behavior, clinical appearance, and treatment options [17,18,19,20,21]. Therefore, these tumors are evaluated using a standardized system that includes tumor staging. In humans, there is a well-established clinicopathological staging proposed by the American Joint Committee on Cancer, evaluating tumor size (T), nodal involvement (N), the presence of distant metastasis (M), and histological grading characteristics (G) [13]. Therefore, in human medicine, the staging system is always used for a better prognosis.

In dogs, a grading system has been applied to classify this group of sarcomas, and a uniform grading system has been proposed for this tumor group [1]; however, no previous staging system or mitotic count has been proven to be associated with patients’ overall survival. Due to the lack of prognostic clinical information for canine STS, this study aimed to assess the prognostic value of pathological and clinical features associated with overall survival time. Furthermore, we propose a new clinicopathological staging system and evaluate a cut-off value for mitosis related to the survival of dogs affected by STS.

## 2. Materials and Methods

### 2.1. Study Design

This retrospective non-randomized multicenter study included 105 dogs with cutaneous and subcutaneous STS. The cases in the present study were retrieved from São Paulo State University-UNESP, Paulista University-UNIP, BICHO e RABICHO Veterinary Clinic, ONCONNECTIONVET, ELETRO-ONKOVET, and SEOVET cancer care oncology services. The inclusion criteria for this study were patients with histopathological analysis confirming STS diagnosis, patients who underwent a full search for metastasis, including thoracic X-ray and abdominal ultrasound, and patients with clinical information available in the respective clinical database. Those excluded from this research were patients who died from causes not related to the STS, and patients who missed follow-up. These exclusion criteria were important because we censored only cases where patients were still alive. Therefore, the survival analysis group was composed only of patients that died from the disease or patients still alive (censored patients). This study was approved by the Committee on Ethics in the Use of Animals (204/2018).

### 2.2. Clinical Data

Clinical data were obtained from patients’ medical records, and inclusion criteria were based on the recommended guidelines for conducting and evaluating prognostic studies in veterinary oncology [22]. Patients with information regarding the involvement of lymph nodes and the presence of distant metastasis, and treated with surgery associated with a free margin (3 cm margin), were included. Additionally, the study exclusively included patients whose regional lymph node assessment was performed by palpation and cytological aspiration, with this information being clearly described in the medical records. The same was applied for distant metastasis, with patients undergoing both a three-view thoracic X-ray and an abdominal ultrasound evaluation. Finally, patients who received any adjuvant or neoadjuvant treatment were excluded.

Regarding follow-up, only patients who died from the STS’s direct involvement were included. For living patients, the minimum acceptable follow-up time was 36 months. It was considered a death related to STSs when the patients died from metastatic disease or due to direct cancer involvement (i.e., massive mass in the thoracic region inducing respiratory distress). Some patients suffering from STSs were euthanized. To ensure the accuracy of our clinical data, these data included only euthanized patients for whom the procedure was performed as a last resort, with no other possibility (i.e., euthanasia performed in a patient with a mass infiltrating the skull two months after splenectomy, with the patient presenting with a seizure non-responsive to medical treatment). Patients who died from other causes were excluded from this study. For tumor localization, the tumors were grouped into those located in the trunk or extremities, according to previous criteria for human STS [23].

### 2.3. Clinicopathological Staging System

The tumor stage was adapted from the human American Joint Committee on Cancer (AJCC) 8th Edition Staging System for Soft Tissue Sarcoma of the Extremity and Trunk [10]. The adapted TNM system included tumor size, lymph node involvement, distant metastasis, and histological grade. The proposed Veterinary TNM system is shown in Table 1, and the new staging system is shown in Table 2. In addition, tumor grade was assessed according to Trojani et al. [23] and Dennis et al. [1].

### 2.4. Morphological Analysis

Hematoxylin and eosin-stained slides were assessed, and a new histological evaluation was performed to confirm the diagnosis by two independent pathologists with experience in STS diagnosis. First, histological analysis was performed at a lower magnification (50×), followed by a subsequent increase in fields (100×, 200×, and 400×). Finally, histological evaluation, classification, and grading were performed according to Dennis et al. [1] and Trojani et al. [23]. Mitotic count was performed using oculars with 22 mm field diameter equaling 2.37 mm^2^ in 10 contiguous fields at 400× magnification.

### 2.5. Statistical Analysis

We investigated the associations among the different study variables using the chi-squared test. Discrete and continuous quantitative variables, such as tumor size and age, were transformed into categories with three groups of an equal number of patients. The variable tumor size was divided into bands of 0–4 cm, 4–7.6 cm, and >7.6 cm; the variable age was divided into 0–8, 8–11.3, and >11.3; and the variable survival in days was divided into 0–120 days, 120–363 days, and >363 days. The results are reported using expected and observed frequencies and Pearson’s chi-square values with corresponding *p*-values.

Additionally, we grouped samples according to clinicopathological criteria and used Kaplan–Meier survival analysis to compare the different categorical variables with overall survival (OS). The considered categories were histological differentiation, mitotic count, percent of necrosis, histological grade, tumor diameter, lymph node metastasis, distant metastasis, clinical stage, and treatment. The mean mitotic count (MC) was 5.7, but considering that the mitotic count presents whole numbers, we considered two groups for mitotic evaluation: patients with ≤5 mitoses and patients with >5 mitoses. In the survival analysis, censoring was only applied to patients who were still alive with no recurrence or distant metastasis.

Statistical analysis was performed using GraphPad Prism v.8.1.0 (GraphPad Software Inc., La Jolla, CA, USA) and the Statistical Package for the Social Sciences (SPSS) program (IBM SPSS Statistics 20.0.0, IBM Corporation, Armonk, NY, USA, 2011). Only results with a significance level of *p* < 0.05 were reported.

## 3. Results

A total of 105 patients met our inclusion criteria and were included in this study. The mean age of these patients was 9.97 (±3.01) years. Mixed-breed dogs were the most represented (53 out of 105), followed by the Poodle (6/105), Golden Retriever (5/105), and Rottweiler (5/105) breeds. Regarding sex, female dogs represented 68 out of 105 patients, and the remaining 37 were male dogs. Among the tumors, 54 out of 105 were located in the trunk, and the remaining 51 were situated in the limbs. The mean size of the tumors was 6.4 (±4.7) cm, and the mean overall survival was 365.91 days (±180 days). Metastasis at diagnosis occurred in 8 out of 105 cases (0.076%).

A significant association between sex and tumor site was found (Pearson’s chi-square = 4.225; *p* = 0.04), with females being more prone to presenting tumors in the trunk than in the extremities (*n* = 40 vs. *n* = 28) and males having a higher prevalence of tumors located in the limbs (*n* = 23 vs. *n* = 14). Additionally, a significant association was found between regional lymph node metastasis and overall survival (Pearson’s chi-square = 6.141; *p* = 0.046). Patients with lymph node metastasis were not expected to survive for more than 120 days. The expected frequencies for the survival bands 0–120, 120–363, and >363 days for the lymph nodes affected were 1.6, 1, and 1.3, while the number of animals observed in each category was 4, 0, and 0, respectively. In contrast, patients without lymph node metastasis were expected to survive for >363 days. The expected frequencies for patients free from lymph node metastasis were 40.4, 26, and 33.7, while the number of animals observed in each category was 38, 27, and 35, respectively.

As for the survival analyses, there was a prevalence of tumors classified as grade II (*n* = 43), followed by I (*n* = 40) and III (*n* = 22). Patients with grade III disease experienced a shorter survival time than those with grade I and II diseases (*p* < 0.001) (Figure 1). Regarding the association of mitotic number with survival, we identified an interesting prognostic association. Patients with tumors presenting ≤5 mitoses had a higher survival time than those with >5 mitoses (*p* = 0.001) (Figure 2).

We identified a shorter survival time for patients with grade III tumors compared with those with grades I and II.

Regarding our proposed clinicopathological stages, the mean overall survival was 445.05 days for patients with stage I, 375.52 days for stage II, 224.21 days for stage III, and 278.50 for stage IV. The patients with tumors presenting with stages III and IV experienced a shorter survival time than those with stages I and II (*p* < 0.0001) (Figure 3).

## 4. Discussion

The main purpose of this study was to identify predictive factors and evaluate a new clinicopathological staging system for dogs affected by STS. In the beginning of the study, our retrospective analysis identified 350 STS patients with available clinical information. However, after the application of exclusion criteria, an important set of patients were disqualified. The major reason for sample exclusion was related to the patient’s death not being associated with disease progression or the inability to locate the owner to confirm the current survival status. These exclusion criteria were applied to make our survival analysis stronger, censoring only patients that we were sure were alive. Therefore, this study included 105 middle-to-older-aged dogs with a female prevalence.

This research identified female dogs as being more prone to presenting tumors in the trunk than in the extremities, whereas, for males, the opposite occurred. We found significant differences between sexes, which suggests that hormones could be associated with STS development. The expression of progesterone receptors was previously associated with peripheral nerve sheath and perivascular wall tumors [24]. Estrogen (ER) and progesterone (PR) receptors are responsible for regulating cell growth and differentiation upon ligand-dependent and independent activation [25]. In human STSs, ER overexpression by immunohistochemistry was previously associated with favorable outcomes in human female patients. On the other hand, PR was considered an unfavorable prognostic marker in male patients [25]. Therefore, it seems that hormone receptor expression could be associated with a patient’s survival. In dogs, ER and PR immunoexpression were assessed; however, no patient’s survival was provided in the study [24]. Therefore, it is not possible to confirm that ER and PR expression are prognostic factors in dogs.

No association was found between tumor localization (trunk or limbs) and clinicopathological factors. These data are in accordance with previous findings, and they suggest the influence of factors independent of the tumor location; these tumors should be grouped as unique entities [17]. The tumor location may influence the surgical margins and complete tumor resection. Additionally, we found that patients with lymph node metastasis were not expected to survive for >120 days, whereas patients without lymph node metastasis were expected to survive for >363 days. It is not surprising that patients with metastatic lymph nodes experienced the shortest survival time. Metastasis in lymph nodes has been associated with poor patient outcome in several cancer subtypes. However, the previous literature was not unanimous in affirming that patients with lymph node metastasis present with a poor prognosis [1,2,3,4,5]. This could be related to the low frequency of lymph node metastasis in patients affected by STSs [26]. Therefore, our study has suggested that patients with lymph node metastasis experienced a shorter survival time, making this a potential prognostic factor. 

Regarding survival analysis, we found that patients with grade III tumors experienced a shorter survival time than those with grade I and II tumors (*p* < 0.001) and that those with tumors presenting ≤5 mitoses experienced a higher survival time (*p* = 0.001). According to the previous literature, the most important prognostic factors are histologic grade and wide surgical margins, which predict no recurrence. In addition, a high mitotic count is considered as a prognostic factor associated with shorter survival time [1]. However, no previous unified mitotic count has been proposed as an independent prognostic factor for STSs. Therefore, the mitotic count is used as part of the tumor grading system. Since our results strongly suggest that mitotic count is an independent prognostic factor for canine STSs, we strongly recommend its use in clinical practice.

Mitotic index is expressed as a percentage of mitotic cells divided by cells not undergoing mitosis [27]. On the other hand, mitotic count is the sum of mitosis in 10 high-power fields. Therefore, some authors have described that over the years, both terms have been mixed and used incorrectly. In this study, we opted to perform the mitotic count because most of the human studies use this parameter to associate it with a patient’s survival [10,28]. Additionally, mitotic count is more often used than mitotic index in Brazilian pathologists’ routines. Therefore, it will be more likely to have been used in the clinical routine. Similar to our results, previous studies have reported a higher frequency of grade I and II tumors, with only a small set of grade III tumors. Although this grading system is determined by histopathological analysis, it has been widely used for the classification of STSs, reinforcing that histological grade is the most crucial prognostic factor for STS [8,19]. Other relevant data in this study are that the lowest survival time was found for grade III tumors compared with that for grade I and II tumors.

The *Cancer Staging Manual* of the AJCC has recently been revised and updated to the eighth edition [29]. Tumor stage is one of the most important prognostic factors for all subtypes of STSs, and we showed a lower survival time in patients with stages IV and III compared with stages I and II. Regarding frequency in this study, stages I (36%) and II (38%) were more frequent than stages III (18%) and IV (7%), not showing a difference from what was proposed in the literature. For canine STSs, no previous clinical stage has been presented. Although we retrospectively enrolled patients in our study, we had a homogenous tumor group, reinforcing the strength of our staging. For human STSs, the clinical stage is one of the most important prognostic factors. Thus, we strongly recommend using the clinicopathological stage for canine STSs.

It is known that STSs have a low metastatic rate (reportedly up to 17% of cases); of the 105 patients included in this study, only seven had metastasis, and two had lymph node infiltration. There was also an important correlation between metastasis and survival rate, which significantly decreased when any metastasis was diagnosed. The local recurrence of marginally excised subcutaneous STSs is variable and difficult to predict. STSs are commonly surrounded by a pseudo capsule that may contain or be confluent with tumor cells [30]. The resemblance to complete margins [31,32] does not predict recurrence. Further research is needed to determine more precise estimates for recurrence and survival rates. Other potential indicators of prognosis that presently require further investigation include histological type, tumor dimension, location, invasiveness, stage, markers of cellular proliferation, and cytogenetic profiles [1,14].

It has been verified that larger tumors are more difficult to remove than smaller ones, and the size of the tumor directly impacts the surgical procedure and the possibility of residual disease. Several canine and human studies have suggested that tumors >5 cm have shorter disease-free intervals or survival times compared with tumors smaller than this size [33,34]. In the respective study, the mean tumor size was 6.4 cm, having a direct correlation with survival time, which was longer in dogs with T < 6.4 cm.

## 5. Conclusions

Our study suggests that the proposed clinicopathological staging and mitotic count present a direct association with the patient’s overall survival. Therefore, we suggest the use of both parameters in clinical practice.

## Figures and Tables

**Figure 1 vetsci-10-00327-f001:**
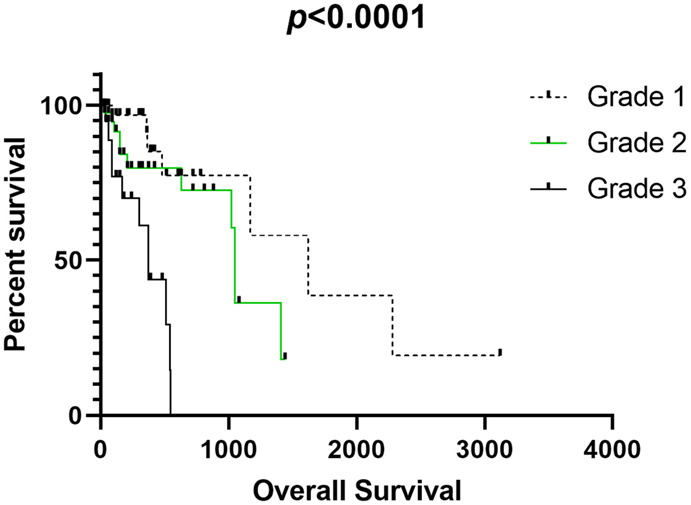
Association of tumor grade with overall survival in our tumor group.

**Figure 2 vetsci-10-00327-f002:**
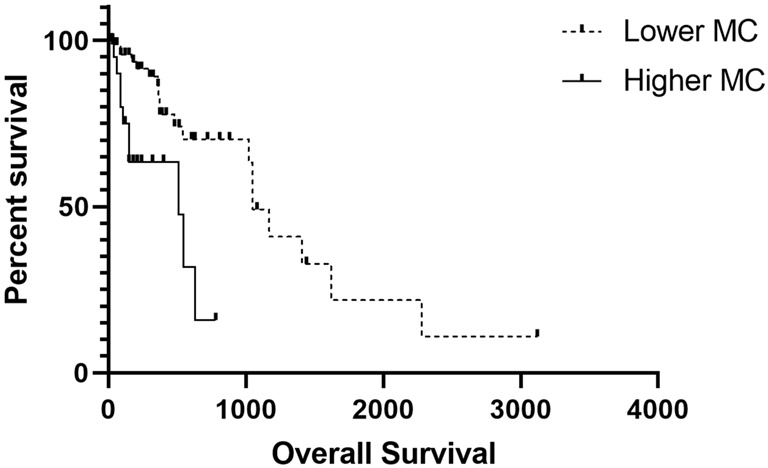
Overall survival association with mitotic count (MC). Patients with tumors presenting > 5 mitoses experienced a shorter survival time.

**Figure 3 vetsci-10-00327-f003:**
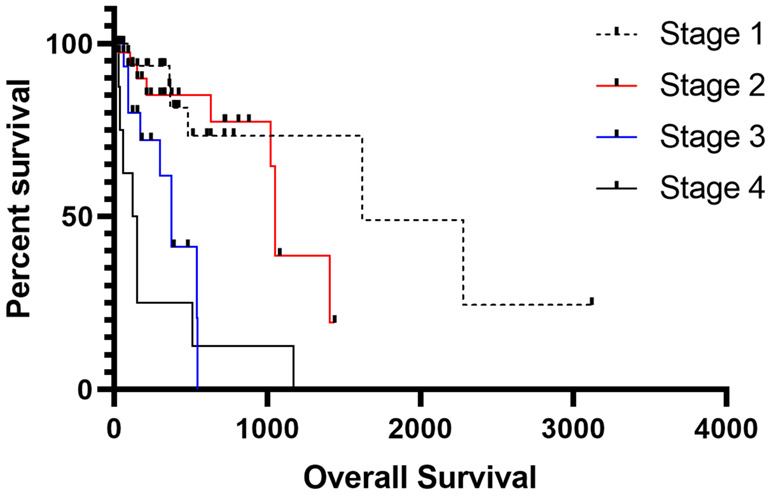
Overall survival associated with the proposed clinicopathological staging system. Patients with stages III and IV experienced a shorter survival time than those with stages I and II.

**Table 1 vetsci-10-00327-t001:** New TNM system for canine cutaneous and subcutaneous soft tissue sarcomas adapted from the human classification system.

Definition of Primary Tumor (T)	
T category	T criteria
Tx	Primary tumors cannot be assessed
T0	No evidence of primary tumor
T1	Tumors 3 cm or less in greatest dimension
T2	Tumors > 3 cm and ≤ 7 cm in the greatest dimension
T3	Tumors > 7 cm and ≤ 12 cm in the greatest dimension
T4	Tumor 15 cm in the greatest dimension
Definition of regional lymph node (N)	
N category	N criteria
N0	No regional lymph node metastasis
N1	Regional lymph node metastasis
Definition of distant metastasis (M)	
M category	M criteria
M0	No distant metastasis
M1	Distant metastasis
Definition of Grade	
Gx	Grade cannot be assessed
G1	Total differentiation with tissue resembling the normal counterpart, mitotic count ≤ 9, no necrosis
G2	Total differentiation with tissue showing poor differentiation, mitotic count between 10 and 19, necrosis ≤ 50%
G3	Undifferentiated sarcoma, mitotic count ≥ 20, necrosis ≥ 50%

**Table 2 vetsci-10-00327-t002:** New clinical staging system proposed for dogs affected by cutaneous and subcutaneous soft tissue sarcomas.

Stage	T	N	M	Histological Grade
Stage I	Tx, T1 or T2	N0	M0	Gx or G1
Stage II	T2, T3 or T4	N0	M0	G2
Stage III	T3 or T4	N0	M0	G3
Stage IV	Any T	N1	M0	Any G
	Any T	N0	M1	Any G

## Data Availability

The data created is already included in the manuscript. Data sharing is not applicable to this article.

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
