# Peer review of "Proposing Clinicopathological Staging and Mitotic Count as Prognostic Factors for Canine Soft Tissue Sarcomas"

_vetsci, 2023, doi:10.3390/vetsci10050327_

Round 1
Reviewer 1 Report
this is a well written and clearly presented report of data obtained from 105 clinical cases of STS confined to the lower limbs in dogs. A clearly stated new staging system was applied to these cases and reported clearly in tables and graphs and included statistical analysis of the data. . It concluded that mitotic count is an independent prognostic indicator and suggested it be used in clinical practice. It was suggested that mitotic numbers( index) be used along with clinico- pathological staging as the best way to predicting survival. As expected a cut off of< 5 mitoses is a better prognostic marker than mitoses > 5. However it is not clearly stated in the methods how many mitotic counts per microscope slide or field size were determined for each tumor. it is noted that only 17% of these STS have mets. Survival was clearly associated with tumor grade and as expected the lower the grade the longer the survival time.(Fig 3) No metastases was also found to be a good prognostic indicator and most important survival factor along with grade of the tumor.
it was commented that local recurrence at the surgery site was difficult to evaluate and predict its value in prognosis.
Author Response
Dear reviewer, thank you so much for your kind overview of our manuscript and for bringing the important point to explain in detail how mitotic count was performed. Mitotic count was performed using oculars with 22mm field diameter equals 2.37mm2 in 10 contiguous fields at 400X magnification. We have included this information in the manuscript. Once again, thank you for the kind overview.
Reviewer 2 Report
Soft tissue sarcomas (STS) account for approximately 15-20% of all cutaneous and subcutaneous tumors in dogs. STS are a heterogeneous set of tumors and are composed of the mesenchymal cells that make up the various forms of connective tissue . STS include fibrosarcoma, hemangiopericytoma, liposarcoma, myxosarcoma, rhabdomyosarcoma, and undifferentiated sarcoma with a different biological progression. Continuous progress in both human and veterinary oncology has shown us that not all cancers are created equal. Therefore, a personalized approach based on the entire constellation of tumor and patient characteristics is more appropriate than a single assignment
Author Response
Dear reviewer, thank you for the kind words and for considering our manuscript well structured and designed. Once again, thank you so much.